# Parental Efficacy in Managing Smartphone Use of Adolescents with Attention-Deficit/Hyperactivity Disorder: Parental and Adolescent Related Factors

**DOI:** 10.3390/ijerph19159505

**Published:** 2022-08-02

**Authors:** Wen-Jiun Chou, Ray C. Hsiao, Cheng-Fang Yen

**Affiliations:** 1Department of Child and Adolescent Psychiatry, Chang Gung Memorial Hospital, Kaohsiung Medical Center, Kaohsiung 83301, Taiwan; wjchouoe2@gmail.com; 2College of Medicine, Chang Gung University, Taoyuan 33302, Taiwan; 3Department of Psychiatry and Behavioral Sciences, University of Washington School of Medicine, Seattle, WA 98295, USA; rhsiao@u.washington.edu; 4Department of Psychiatry, Seattle Children’s, Seattle, WA 98105, USA; 5Department of Psychiatry, Kaohsiung Medical University Hospital, Kaohsiung 80708, Taiwan; 6Department of Psychiatry, School of Medicine, Kaohsiung Medical University, Kaohsiung 80708, Taiwan; 7College of Professional Studies, National Pingtung University of Science and Technology, Pingtung 91201, Taiwan

**Keywords:** parental efficacy, attention-deficit/hyperactivity disorder, smartphone use, parenting, depressive symptoms, oppositional defiant disorder, adolescents

## Abstract

Parental management has an important role in preventing problematic smartphone use among adolescents with attention-deficit/hyperactivity disorder (ADHD). This study aimed to examine the parental factors (e.g., demographics, depressive symptoms and parenting styles) and adolescent factors (e.g., demographics, ADHD and oppositional defiant disorder [ODD] symptoms, and problematic smartphone use) related to parental efficacy in managing adolescent smartphone use (PEMASU) among 237 parents of adolescents with ADHD. PEMASU was measured by the Parental Smartphone Use Management Scale. Parental depressive symptoms and parenting styles (parental affection/care and overprotection) were measured by the Center for Epidemiologic Studies—Depression Scale and Parental Bonding Instrument, respectively. Adolescent ADHD and ODD symptoms and problematic smartphone use were measured by the Swanson, Nolan, and Pelham, version IV scale and Problematic Smartphone Use Questionnaire, respectively. Three models of hierarchical linear regression were performed to examine the parental and adolescent factors related to PEMASU. The results indicated that adolescent older age and more severe ODD symptoms and problematic smartphone use were significantly associated with lower PEMASU, whereas greater parental affection/care was significantly associated with higher PEMASU. This study demonstrated that both parental and adolescent factors contribute to PEMASU among parents of adolescents with ADHD. Intervention programs aiming to enhancing PEMASU need to take these related factors into consideration.

## 1. Introduction

### 1.1. Problematic Smartphone Use among Adolescents with Attention-Deficit/Hyperactivity Disorder (ADHD)

Problematic smartphone use (or smartphone addiction) is common among adolescents with ADHD. A study conducted in 4512 South Korean adolescents demonstrated that ADHD symptoms were the most significant factor related to the severity of problematic smartphone use [1]. Another study on middle high school students in South Korea also demonstrated that the severity of ADHD symptoms was significantly associated with smartphone addiction [2]. Multiple biopsychosocial characteristics may contribute to the high risk of problematic internet/smartphone use in adolescents with ADHD, including proneness to boredom, aversion to delayed rewards, difficulties in controlling impulsivity and social adaptation, and rapid habituation to repeated positive reinforcement [3,4,5]. Longitudinal studies have determined that problematic smartphone use predicts the development of negative psychological consequences such as low self-esteem [4], loneliness [6], and depression [6] in children and adolescents; thus, problematic smartphone use in adolescents with ADHD warrants early prevention, identification, and intervention.

### 1.2. Parental Efficacy in Managing Smartphone Use of Adolescents with ADHD

The studies examining family factors of problematic smartphone use in adolescents are scarce [5]. Previous studies have found that parental smartphone addiction [7,8], poor parent–child relationship and communication [7,8,9,10,11,12,13], and low parental support [14] increase the risk of problematic smartphone use. Parenting styles are one of family factors that have received researchers’ attention. A democratic parenting style can protect children and adolescents from problematic smartphone use [15], whereas overprotective [16,17] and strict parenting [5] can increase the likelihood of problematic smartphone use. However, another study did not find a significant association between parenting styles and adolescent smartphone addiction [2]. Moreover, the indirect effect of parental smartphone addiction on adolescent smartphone addiction through parent–child relationship was greater in the families with higher overprotective parenting than in those with lower overprotective parenting [7].

Parental efficacy in managing adolescent smartphone use (PEMASU) may affect whether parents can successfully carry out the management for adolescent problematic smartphone use; however, this has not been discussed in depth in the literature. Overall parental efficacy refers to the belief and confidence the parents have in their competence to successfully execute parenting behaviors and foster children’s positive development [18]. Parents who have a high overall efficacy can create adaptive child-rearing environments [19] and reducing the risk of developing problematic behaviors in their children [18]. Research has found that adolescents with ADHD conflicted with their parents more frequently compared with those without ADHD [20,21,22]. Moreover, compared with parents of children without ADHD, parents of children with ADHD reported a lower level of overall parental efficacy [23].

Situation-specific parental efficacy refers to parents’ belief and confidence the parents have in their competence to ensure children’s attainments of some specific tasks in their daily lives [24]; for example, supervising children’s toothbrushing [25] and coping with natural and human-made disasters [26]. Hsieh et al. [27] firstly developed the Parental Smartphone Use Management Scale and proposed three core dimensions of PEMASU for parents of adolescents with ADHD. The first dimension, “reactive management”, indicates parents’ efficacy in managing ADHD adolescents’ smartphone use through setting and practicing rules and in avoiding the negative impacts of smartphone use on ADHD adolescents’ daily life functioning. Research has found that longer duration of smartphone use on a typical day [28,29] and a shorter time period until first smartphone use in the morning [28] were significantly associated with adolescent smartphone addiction. The results supported the importance of parental self-efficacy in setting and practicing rules for adolescent smartphone use. The second dimension, “proactive management,” indicates parents’ efficacy in managing ADHD adolescents’ smartphone use through positive communication and reasoning. Previous studies have identified poor parent–child relationship and communication as a risk factor of adolescent smartphone addiction [7,8,9,10,11,12,13] and thus supported the necessity of parental self-efficacy in well communicating with adolescents with ADHD regarding smartphone use. The third dimension, “monitoring”, indicates parents’ efficacy in monitoring what ADHD adolescents do on smartphones, whom they talk to, and what applications and websites they use and visit [27]. Previous studies have found that using a smartphone for gaming [5,28,29], social networking [2,5,28,29], and music/videos [2,29] were risk factors of smartphone addiction in adolescents. The results further supported the importance of parental self-efficacy in monitoring the contents of adolescent smartphone use. Mental health and educational professionals should consider and enhance these dimensions of parental efficacy in order to reduce the risk of problematic smartphone use in adolescents with ADHD [27].

### 1.3. Factors Related to Parental Efficacy in Managing Smartphone Use of Adolescents with ADHD

Factors that relate to PEMASU in parents of adolescents with ADHD have not been surveyed. As observed in the socioecological model [30], both adolescent and parental factors may influence the level of PEMASU in parents of adolescents with ADHD. First, the severity of adolescent problematic smartphone use may have a bidirectional relationship with PEMASU. Severe adolescent problematic smartphone use may increase the difficulties faced by parents in managing the behavior and then compromise PEMASU; alternatively, low PEMASU may lead to parental failure to adequately manage their children’s problematic smartphone use. Second, ADHD symptoms were significantly associated with the difficulties in self-control [31] that may hamper the self-control of smartphone use [32]. More severe ADHD symptoms may also have less cooperation with parental management owing to distraction and impulsivity, consequently decreasing PEMASU. Third, as one of the common comorbid externalizing behavioral problems in children and adolescents with ADHD [33], the symptoms of oppositional defiant disorder (ODD) predicted negative parental management through decreased parental efficacy [34]. Strong ODD behavior may lead to parents’ frustrated feelings and low confidence in parenting. Fourth, regarding adolescent demographics, the results of previous studies on the roles of gender and age in the risk of adolescent smartphone addiction were mixed [5]. Regarding parental factors, research reported that parental depression was negatively associated with parental efficacy [35]. Moreover, various parenting styles have different relationships with the risk of problematic smartphone use in children and adolescents [5,15,16,17]. Affectionate and caring parenting behaviors have been considered a positive parenting style that can result in good parent–child interaction, whereas overprotective parenting may not be helpful to parent–child interaction [36]. Research has found that parents who have high self-efficacy are more likely to display affectionate and caring parenting behaviors and adopt an authoritative parenting style [37,38]. However, whether these parental and adolescent factors are significantly associated with PEMASU warrants further study.

### 1.4. Aims of This Study

Through this study, we aimed to examine the associations of parental factors (e.g., demographics, depressive symptoms, and parenting styles) and adolescent factors (e.g., demographics, ADHD and ODD symptoms, and problematic smartphone use) with PEMASU in parents of adolescents with ADHD. We hypothesized that more severe adolescent problematic smartphone use, adolescent ADHD and ODD symptoms and parental depression, and greater overprotective parenting style were significantly associated with lower PEMASU in parents of adolescents with ADHD, whereas a democratic parenting style was significantly associated with higher PEMASU.

## 2. Materials and Methods

### 2.1. Participants and Procedure

The methods of recruiting participants have been described elsewhere [27]. In brief, 237 parents of adolescents with ADHD were consecutively recruited from 8 child psychiatric outpatient clinics of two hospitals in Kaohsiung, Taiwan during the period between August 2014 and July 2015. The inclusion criterion was parents who had any child aged between 11 and 18 years old who had a diagnosis of ADHD according to the Diagnostic and Statistical Manual of Mental Disorders, Fifth Edition [39]. The exclusion criteria were as follows: (1) Parents of ADHD adolescents having comorbid psychiatric disorders (e.g., intellectual disability, autism spectrum disorder, language disorder, schizophrenia, bipolar I disorder, and major depressive disorder) that resulted in considerable difficulties in communication, emotional expression, and behavioral control; and (2) parents with any cognitive impairment (e.g., neurodevelopmental disorder, substance use disorder, intellectual disability, autism spectrum disorder, schizophrenia, and bipolar I disorder) that may have impaired their ability to understand the purpose and procedure of the study.

Two child psychiatrists (WJC and CFY) confirmed the eligibility of 237 potential participants, and 231 (97.5%) parents agreed to participate. Regarding the demographics of participants, 192 were mothers and 39 were fathers; mean age was 43.8 years (standard deviation [SD] = 6.1 years); mean number of years of academic education completed was 13.8 years (SD = 2.8 years). Regarding the demographics of adolescents with ADHD, 32 were girls and 199 were boys; mean age was 13.7 years (SD = 1.8 years); 190 were in a two-parent family.

All participants provided written informed consent. Trained research assistants explained to the participants how to complete the research questionnaires and assured that their responses would be confidential. Participants completed the research questionnaires in the interview rooms of the psychiatric outpatient units individually and could seek help from the research assistants if they experienced difficulties in completing the questionnaire. The Institutional Review Board of Kaohsiung Medical University approved the study (approval number: KMUHIRB-20130131; date of approval: 11 February 2014).

### 2.2. Measures

#### 2.2.1. Parental Smartphone Use Management Scale (PSUMS)

The PSUMS contained 17 items assessing the 3 dimensions of PEMASU (reactive management, proactive management, and parental monitoring) in the previous one month [27]. Participants rated each item on a 7-point scale ranging from 0 (*no efficacy at all*) to 6 (*extremely strong efficacy*). Participants who rated a higher mean score had a higher level of PEMASU. A previous study on parents of adolescents with ADHD supported that the PSUMS had acceptable reliability and validity [27]. The internal consistency of the PSUMS was high in the present study (Cronbach’s α = 0.97).

#### 2.2.2. Chinese Version of the Swanson, Nolan, and Pelham Version IV Scale (C-SNAP-IV)—Parent Form

The C-SNAP-IV–parent form contained 26 items assessing children’s inattention (9 items), hyperactivity/impulsivity (9 items) and ODD symptoms (8 items) in the previous one month [40,41]. Participants rated each item on a 4-point scale ranging from 0 (*not at all*) to 3 (*very much*). Participants who rated a higher mean score had the children with more severe ADHD and ODD symptoms. Previous studies have confirmed that the C-SNAP-IV had acceptable reliability and validity [40]. The internal consistency of the 3 dimensions of the C-SNAP-IV was high in the present study (Cronbach’s α of the inattention, hyperactivity/impulsivity, and ODD subscales = 0.91, 0.90, and 0.93, respectively).

#### 2.2.3. Problematic Smartphone Use Questionnaire (PSUQ)

The PSUQ contained 8 items assessing the severity of children’s problematic smartphone use in the previous one month [42]. Participants rated each item on a 4-point scale ranging from 1 (*very unlikely*) to 4 (*very likely*). Participants who rated a higher mean score had the children with more severe problematic smartphone use. A previous study demonstrated that the PSUQ had acceptable psychometric propensities [42].

#### 2.2.4. Mandarin Version of the Center for Epidemiologic Studies—Depression Scale (M-CES-D)

The M-CES-D contained 20 items assessing the frequency of parental depressive symptoms in the previous one month [43,44]. Participants rated each item on a 4-point scale ranging from 0 (*never*) to 3 (*always*). A previous study found that the M-CES-D had acceptable psychometric propensities [43]. The internal consistency of the M-CES-D was acceptable in the present study (Cronbach’s α = 0.81).

#### 2.2.5. Chinese Version of the Parental Bonding Instrument (C-PBI)

The C-PBI contained 25 items assessing 2 dimensions of long-term parenting behaviors (parental affection/care and overprotection) [20,45]. Participants rated each item on a 4-point scale ranging from 1 (*v**ery likely*) to 4 (*v**ery unlikely*). Participants who rated a higher mean score on the parental affection/care and overprotection dimensions had greater affectionate and warm parenting behaviors and denial of adolescent psychological autonomy with discouragement of adolescent behavioral freedom, respectively. A previous study found that the C-PBI has acceptable reliability and validity [20]. The internal consistency of the parental affection/care and overprotection on the C-PBI was acceptable in the present study (Cronbach’s α = 0.74 and 0.74, respectively).

### 2.3. Statistical Analysis

Categorical and continuous variables were first analyzed using frequency (percentage) and mean (SD), respectively. The correlations among the variables were examined using Pearson correlation analyses. Three models of hierarchical linear regression were used to test the associations of parental and adolescent factors with PEMASU. The 3 sets of independent variables were demographic variables (parents’ sex and education, and adolescents’ sex and age), adolescent factors (adolescent ADHD and ODD symptoms and problematic smartphone use), and parental factors (parents’ depressive symptoms, parental affection/care, and overprotection). Parents’ age was not considered in the analysis because of its significant relationships with other demographic variables. The adjusted R^2^ was included in determining the percentage of variance explained by each model.

## 3. Results

The results of descriptive statistics and correlation coefficients for the variables are presented in Table 1. The mean score of the PSUMS was 4.05, indicating an upper level of PEMASU. The mean item scores of inattention and ODD symptoms were 1.41 and 1.23, respectively, indicating a mild to moderate level of inattention and ODD symptoms. The mean item score of hyperactivity/impulsivity was 0.97, indicating a mild level of hyperactivity/impulsivity. The mean item scores of parental affection/care and overprotection were 2.87 and 2.32, respectively, indicating a mild to moderate tendency of parental affection/care and overprotection. The mean item score of the M-CES-D was 0.73, indicating a low severity of parental depression. The mean item score of the PSUQ was 2.32, indicating a mild to moderate level of adolescent problematic smartphone use. Adolescents’ age, ODD and inattention symptoms and problematic smartphone use, and parental depression were negatively associated with PEMASU, whereas parental education and parental affection/care were positively associated with PEMASU. Parents’ sex and age and adolescents’ sex were not significantly associated with PEMASU.

The results of three hierarchical linear regression models examining the associations of parental and adolescent factors with PEMASU are shown in Table 2. The result of Model 1 revealed that older age of adolescents was associated with lower PEMASU. Adolescent factors were further selected in Model 2, which indicated that after controlling for the effects of demographic factors, more severe adolescent ODD symptoms and problematic smartphone use were significantly associated with lower PEMASU, but the effect of age became nonsignificant. Parental factors were further added in Model 3, which indicated that after controlling for the effects of demographic and adolescent factors, greater parental affection/care was significantly associated with higher PEMASU, and the effects of adolescent ODD symptoms and problematic smartphone use remained significant. We did not discover significant associations between PEMASU and variables such as adolescents’ sex, parents’ sex and years of education, adolescent hyperactivity/impulsivity and inattention symptoms, parental depressive symptoms, and parental overprotection. The final model explained 22% of the variance.

## 4. Discussion

### 4.1. Adolescent Factors Related to PEMASU

The present study demonstrated that the severities of ODD symptoms and problematic smartphone use in adolescents with ADHD were negatively associated with PEMASU. Core ODD symptoms include anger, irritable mood, argumentativeness, defiance, and vindictiveness [39]. Adolescents with severe ODD symptoms often argue with people in authority and actively refuse to comply with parents’ requests. This defiance may increase parents’ difficulties in managing adolescents’ smartphone use behaviors. Adolescents with severe ODD symptoms may also react to parents’ management with outrage and resentfulness, which may annoy the parents and other family members. All these behavioral and emotional reactions to parental management may further reduce PEMASU.

Adolescents who have problematic smartphone use may increase the time spent on using smartphone to achieve satisfaction [28,29], become preoccupied with smartphone activities such as texting and gaming [2,5,28,29], and become restless and angry when they are unable to use their smartphone [46]. Severe problematic smartphone use indicates a high tendency of resistance to changing smartphone use behaviors. Therefore, severe problematic smartphone use can increase the challenges for parents to manage adolescent smartphone use and compromise PEMASU. Alternatively, low PEMASU may reciprocally increase adolescents’ noncompliance with parental management and further worsen problematic smartphone use. Although the cross-sectional design of this study limited the inference to determine the causal relationship between problematic smartphone use and PEMASU, parents of ADHD adolescents with problematic smartphone use should develop adequate management skills rather than simply applying restrictions on and punishment for adolescent smartphone use. Research demonstrated that behavior-based educational programs based on the principle of cognitive behavioral therapy for both adolescents and parents can improve the severity of adolescent problematic smartphone use [47]. Mental health professionals should actively invite parents of adolescents with ADHD to join such treatment programs.

The present study found that older age of adolescents was associated with lower PEMASU; however, the effect of age became nonsignificant when the effects of ODD symptoms and problematic smartphone use were considered. Presentations of ODD symptoms and problematic smartphone use may become more severe and intense as adolescents grow up. This study did not discover significant associations of inattention and hyperactivity/impulsivity with PEMASU. The participants in this study were invited from clinical units; therefore, their children received pharmacological and/or behavioral treatment for ADHD. ADHD symptoms might be alleviated.

### 4.2. Parental Factors Related to PEMASU

The present study demonstrated that a parenting style of more affection/care was significantly associated with greater PEMASU in parents of adolescents with ADHD. Affectionate and caring parenting behaviors can result in good parent–child interaction [36] that can serve as the basis of parent–child communication and reaching a consensus for adolescent smartphone use. Alternatively, parents who have high PEMASU may have the confidence to adopt an authoritative parenting style and patiently communicate with children with a firm attitude [37,38]; affectionate and caring parenting behaviors were also displayed simultaneously. Therefore, parental affection/care and PEMASU might have reciprocal relationship with each other. Based on the result, enhancing affectionate and caring parenting behaviors may be a means of increasing PEMASU and parents’ competence in managing smartphone use of adolescents with ADHD in the digital era. However, given that parenting behaviors involve long-term interactions between parents and their children, low parental affection/care and low PEMASU may both be the result of frustration due to repeated failure in managing ADHD adolescents’ problematic smartphone use.

### 4.3. Limitations and Further Study

To the best of our knowledge, this study is the first to examine the parental and adolescent factors related to PEMASU among parents of adolescents with ADHD. However, we collected the data based on the parents’ self-report; therefore, there might be single-rater bias. Future studies should collect data from adolescents and not only from parents to reduce the single-rater bias. Second, this cross-sectional study could not make the inference in the temporal relationships between the related factors and PEMASU. Further prospective studies are needed to examine the temporal relationships. Third, the participants were recruited from outpatient clinic units. The source of participants limited the generalization of the results found in this study to parents and ADHD adolescents who have not visited clinic units.

The association between ADHD symptoms and PEMASU should also be investigated in the population of ADHD adolescents who have not visited a clinical unit. Several models of behavioral treatment for ODD problems have been proposed for adolescents with ADHD [48,49]. Whether the behavior-based educational programs can improve smartphone use behaviors of adolescents with ODD and PEMASU warrants further investigation.

## 5. Conclusions

The present study identified that ADHD adolescents’ ODD symptoms and problematic smartphone use were negatively associated with PEMASU, whereas parental affection/care was positively associated with PEMASU in parents of adolescents with ADHD. The results indicate that both parental and adolescent factors correlated with PEMASU in parents of adolescents with ADHD. The present study highlights the need for family-based interventions but not only focusing on adolescents for problematic smartphone use in adolescents with ADHD. Provision of parenting resources to enhance PEMASU and affectionate and caring parenting is warranted to increase PEMASU among parents of adolescents with ADHD and reduce the risk of adolescent problematic smartphone use in this digital age.

## Figures and Tables

**Table 1 ijerph-19-09505-t001:** Bivariate Correlations, Means, and Standard Deviations for the Variables.

	1	2	3	4	5	6	7	8	9	10
1. PEMASU	--									
2. Adolescents’ age	−0.20 **	--								
3. Parents’ years of education	0.16 *	−0.08	--							
4. Adolescents’ ODD symptoms	−0.27 ***	0.13	−0.14 *	--						
5. Adolescents’ H/I symptoms	−0.09	−0.04	−0.10	0.65 ***	--					
6. Adolescents’ inattention symptoms	−0.15 *	0.05	0.01	0.54 ***	0.63 ***	--				
7. Parental depressive symptoms	−0.18 **	0.09	−0.28 **	0.35 ***	0.25 ***	0.21 **	--			
8. Parental affection/care	0.29 ***	−0.01	0.17	−0.09	−0.10	−0.05	−0.23 ***	--		
9. Parental overprotection	−0.11	−0.18 **	−0.11	0.07	0.20 **	0.07	0.13	−0.29 ***	--	
10. Adolescents’ problematic smartphone use	−0.39 ***	0.20 **	−0.18 *	0.38 ***	0.18 **	0.33 ***	0.28 ***	−0.20 **	0.16 *	--
*Scale Range*	0−6	11−18	6−28	0−23	0−3	0−3	0−2.40	1.77−3.69	1.42−3.58	1−4
*Mean*	4.05	13.73	13.51	1.23	0.97	1.41	0.73	2.87	2.27	2.32
*SD*	1.26	1.82	2.80	0.72	0.67	0.67	0.47	0.38	0.39	0.70

PEMASU: Parental efficacy in managing adolescent smartphone use; ODD: Oppositional defiant disorder; H/I: Hyperactivity/impulsivity; *SD:* Standard deviation. * *p* < 0.05; ** *p* < 0.01; *** *p* < 0.001.

**Table 2 ijerph-19-09505-t002:** Factors Related to Parental Efficacy in Managing Adolescent Smartphone Use: Hierarchical Regression Analysis of.

	Model 1			Model 2			Model 3		
Variables	*B*	*SE*	*β*	*B*	*SE*	*β*	*B*	*SE*	*β*
**Step 1: Demographic**									
Parents’ sex	0.239	0.232	0.071	0.107	0.218	0.032	0.197	0.216	0.059
Parents’ years of education	0.058	0.031	0.130	0.031	0.030	0.069	0.018	0.030	0.039
Adolescents’ sex	0.078	0.253	0.021	0.211	0.240	0.057	0.274	0.236	0.074
Adolescents’ age	−0.121	0.047	−0.176 *	−0.082	0.045	−0.119	−0.085	0.044	−0.124
**Step 2: Adolescents’ factors**									
Adolescents’ ODD symptoms				−0.040	0.020	−0.182 *	−0.042	0.019	−0.190 *
Adolescents’ inattention symptoms				0.000	0.018	−0.001	−0.003	0.017	−0.015
Adolescents’ H/I symptoms				0.016	0.020	0.076	0.019	0.020	0.090
Adolescents’ problematic smartphone use				−0.071	0.016	−0.311 ***	−0.061	0.016	−0.270 ***
**Step 3: Parental factors**									
Parents’ depressive symptom							0.005	0.010	0.034
Parental affection/care							0.060	0.017	0.234 ***
Parental overprotection							−0.004	0.024	−0.012
*Adjusted R^2^*	0.05			0.18			0.22		

H/I: Hyperactivity/impulsivity; ODD: Oppositional defiant disorder. * *p* < 0.05; *** *p* < 0.001.

## Data Availability

The data used in this study are available upon reasonable request to the corresponding authors.

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
