# Peer review of "Parental Efficacy in Managing Smartphone Use of Adolescents with Attention-Deficit/Hyperactivity Disorder: Parental and Adolescent Related Factors"

_ijerph, 2022, doi:10.3390/ijerph19159505_

Round 1

Reviewer 1 Report

The manuscript Parental Efficacy in Managing Smartphone Use of Adolescents with Attention-Deficit/Hyperactivity Disorder: Parental and 3 Adolescent Related Factors reviews a cross-sectional study that examines the relationships of parental and adolescent factors in relation to parental efficacy in managing smartphone use of adolescents with ADHD. The subject matter, the study and the findings presented in the manuscript are an important contribution to understanding parental efficacy in managing adolescent smartphone use (PEMASU). However, in order to present the study and its findings clearly and in-depth, the manuscript needs some additional work, as suggested below:

1.     The literature review presented in the Introduction could be more thorough and in-depth. This includes for example adding more information and explaining why the subject matter of the references is important and how the material connects to the content of the study presented. Some of the references mentioned in the discussion should be first mentioned in the literature review and then revisited in the conclusion.

2.     There is some unnecessary, obvious and general information in the literature review that can be removed:

L. 53-57

“Even so, parents are still the principal caregivers who take care of and monitor the behaviors of adolescents with ADHD. Parental efficacy plays an essential role in successfully managing adolescents’ behavioral problems and fostering adolescents’ good behaviors. Parental efficacy can be overall or situation specific.”

3.     There is also some unnecessary general information provided in section 2.3 which can be removed.

L. 177

“We analyzed the data using SPSS software version 25 (SPSS Inc., Chicago, IL, USA).”

L. 188

“A value of p < 0.05 was considered statistically significant.”

4.     For this paragraph in the results section, L. 190-199, I suggest a more in-depth analysis. For example, why is this an important finding and how is it linked to the aims of the study? Are those numbers high or low, and in relation to what?

5.     The discussion section needs sorting out.

a.     The suggestion about further research should be in the same place in section 4.3. Such as these sentences:

L. 234-235: “Whether these interventions can improve smartphone use behaviors of adolescents with ODD and PEMASU warrants further investigation.” And L. 261-263: “The association between ADHD symptoms and PEMASU should also be investigated in the population of ADHD adolescents who have not visited a clinical unit.”

b. Also, make sure that nothing new is brought up in the discussion section. The references mentioned in the discussion section should be first mentioned in the literature review, then they should be revisited in the discussion section in the context of the findings of the study.

In summary, I suggest the following:

1.     On the whole, some parts of the manuscript lack in-depth discussion. That is, parts of it read more like a report rather than a manuscript.  References, the study itself, its findings and in-depth discussion should be connected. Also, removing some general material is needed.

2.     In terms of the line of thought in the article: one of the weaknesses is the organization of the material located at the beginning and end of the manuscript.

·      There needs to be a more in-depth discussion in the introduction about the research and related references, which then would be followed through in the investigation and then revisited in the discussion and conclusion sections.

·      Mention of new references and new points of view should be avoided in the discussion and conclusion sections. Only points that have been worked with during the investigation and mentioned earlier, and which are part of the context of the research, should be included in the discussion and conclusion sections.

·      Some of the material and references could be moved from the discussion section into the introduction. This would provide a better introduction to the study and make the line of thought more logical. Some of those points could be revisited in the discussion section.

Author Response

We appreciated your valuable comments. As discussed below, we have revised our manuscript with underlines based on your suggestions. Please let us know if we need to provide anything else regarding this revision.

Comment

  1. The literature review presented in the Introduction could be more thorough and in-depth. This includes for example adding more information and explaining why the subject matter of the references is important and how the material connects to the content of the study presented. Some of the references mentioned in the discussion should be first mentioned in the literature review and then revisited in the conclusion.

Response

Thank you for your suggestion.

1a. In the revised manuscript we added more introductions regarding problematic smartphone use in adolescents, parental factors related to adolescent problematic smartphone use, and parental self-efficacy in managing adolescent smartphone use into Introduction section. For example,

Another study on middle high school students in South Korea also demonstrated that the severity of ADHD symptoms was significantly associated with smartphone addiction [2]. Multiple biopsychosocial characteristics may contribute to the high risk of problematic internet/smartphone use in adolescents with ADHD, including proneness to boredom, aversion to delayed rewards, difficulties in controlling impulsivity and social adaptation, and rapid habituation to repeated positive reinforcement [35]. Please refer to line 45-51.

The studies examining family factors of problematic smartphone use in adolescents are scarce [5]. Previous studies have found that parental smartphone addiction [7,8], poor parent-child relationship and communication [7−13], and low parental support [14] increase the risk of problematic smartphone use. Parenting styles are one of family factors that have received researchers’ attention. A democratic parenting style can protect children and adolescents from problematic smartphone use [15], whereas overprotective [16,17] and strict parenting [5] can increase the likelihood of problematic smartphone use. However, another study did not find a significant association between parenting styles and adolescent smartphone addiction [2]. Moreover, the indirect effect of parental smartphone addiction on adolescent smartphone addiction through parent-child relationship was greater in the families with higher overprotective parenting than in those with lower overprotective parenting [7]….Parental efficacy in managing adolescent smartphone use (PEMASU) may affect whether parents can successfully carry out the management for adolescent problematic smartphone use; however, it has not being deeply discussed. Please refer to line 57-71.

Research has found that longer duration of smartphone use on a typical day [28,29] and a shorter time period until first smartphone use in the morning [28] were significantly associated with adolescent smartphone addiction. The results supported the importance of parental self-efficacy in setting and practicing rules for adolescent smartphone use. ...Previous studies have identified poor parent-child relationship and communication as a risk factor of adolescent smartphone addiction [7−13] and thus supported the necessity of parental self-efficacy in well communicating with adolescents with ADHD regarding smartphone use. Previous studies have identified poor parent-child relationship and communication as a risk factor of adolescent smartphone addiction [7−13] and thus supported the necessity of parental self-efficacy in well communicating with adolescents with ADHD regarding smartphone use….Previous studies have found that using a smartphone for gaming [5,28,29], social networking [2,5,28,29], and music/videos [2,29] were risk factors of smartphone addiction in adolescents. The results further supported the importance of parental self-efficacy in monitoring the contents of adolescent smartphone use.Please refer to line 88-103.

Fourth, regarding to adolescent demographics, the results of previous studies on the roles of gender and age in the risk of adolescent smartphone addiction were in mixed [5]. ... Moreover, various parenting styles have different relationships with the risk of problematic smartphone use  in children and adolescents [5,1517]. Affectionate and caring parenting behaviors has been considered as a positive parenting style that can result in good parent-child interaction, whereas overprotective parenting may not be helpful to parent-child interaction [36]. Research has found that parents who have high self-efficacy are more likely to display affectionate and caring parenting behaviors and adopt an authoritative parenting style [37,38]. Please refer to line 124-134.

1b. We also moved most of the references mentioned in the discussion into the literature review. For example, original reference 30 was changed to reference 33 (please refer to line 121); original reference 35 was changed to reference 36 (please refer to line 132); original reference 36 was changed to reference 37 (please refer to line 134); original reference 37 was changed to reference 38 (please refer to line 134).

Comment

  1. There is some unnecessary, obvious and general information in the literature review that can be removed: Line 53-57

“Even so, parents are still the principal caregivers who take care of and monitor the behaviors of adolescents with ADHD. Parental efficacy plays an essential role in successfully managing adolescents’ behavioral problems and fostering adolescents’ good behaviors. Parental efficacy can be overall or situation specific.”

Response

Thank you for your comment. We deleted these sentences from the revised manuscript. Please refer to line 71.

Comment

  1. There is also some unnecessary general information provided in section 2.3 which can be removed. Line 177

“We analyzed the data using SPSS software version 25 (SPSS Inc., Chicago, IL, USA).” Line 188

“A value of p < 0.05 was considered statistically significant.”

Response

Thank you for your comment. We deleted these sentences from the revised manuscript. Please refer to line 221 and 231.

Comment

  1. For this paragraph in the results section, L. 190-199, I suggest a more in-depth analysis. For example, why is this an important finding and how is it linked to the aims of the study? Are those numbers high or low, and in relation to what?

Response

Thank you for your comment. We added explanations for the findings of parental and adolescent factors including parental self-efficacy in managing adolescent smartphone use, depression, and parenting styles as well as adolescent problematic smartphone use, ADHD and ODD symptoms. Please refer to line 234-242.

The mean score of the PSUMS was 4.05, indicating a upper level of PEMASU. The mean item scores of inattention and ODD symptoms were 1.41 and 1.23, respectively, indicating a mild to moderate level of inattention and ODD symptoms. The mean item score of hyperactivity/impulsivity was 0.97, indicating a mild level of hyperactivity/impulsivity. The mean item scores of parental affection/care and overprotection were 2.87 and 2.32, respectively, indicating a mild to moderate tendency of parental affection/care and overprotection. The mean item score of the M-CES-D was 0.73, indicating a low severity of parental depression. The mean item score of the PSUQ was 2.32, indicating a mild to moderate level of adolescent problematic smartphone use.

Comment

  1. The discussion section needs sorting out.

5a.     The suggestion about further research should be in the same place in section 4.3. Such as these sentences:

  1. 234-235: “Whether these interventions can improve smartphone use behaviors of adolescents with ODD and PEMASU warrants further investigation.” And L. 261-263: “The association between ADHD symptoms and PEMASU should also be investigated in the population of ADHD adolescents who have not visited a clinical unit.”

Response

Thank you for your suggestion. We added a paragraph in section 4.3. titled “Directions of Further Study” to contain the sentences below. Please refer to line 322 and 333-338.

“4.3. …Directions of Further Study”

The association between ADHD symptoms and PEMASU should also be investigated in the population of ADHD adolescents who have not visited a clinical unit. Several models of behavioral treatment for ODD problems have been proposed for adolescents with ADHD [48,49]. Whether the behavior-based educational programs can improve smartphone use behaviors of adolescents with ODD and PEMASU warrants further investigation.

Comment

5b. Also, make sure that nothing new is brought up in the discussion section. The references mentioned in the discussion section should be first mentioned in the literature review, then they should be revisited in the discussion section in the context of the findings of the study.

Response

We moved most of the references mentioned in the discussion into the literature review. For example, original reference 30 was changed to reference 33 (please refer to line 121); original reference 35 was changed to reference 36 (please refer to line 132); original reference 36 was changed to reference 37 (please refer to line 134); original reference 37 was changed to reference 38 (please refer to line 134).

Comment 

In summary, I suggest the following:

6a. On the whole, some parts of the manuscript lack in-depth discussion. That is, parts of it read more like a report rather than a manuscript.  References, the study itself, its findings and in-depth discussion should be connected. Also, removing some general material is needed.

Response

We revised the manuscript accordingly. Please refer to the response to Comments 1 to 3.

Comment

  1. In terms of the line of thought in the article: one of the weaknesses is the organization of the material located at the beginning and end of the manuscript.

7a. There needs to be a more in-depth discussion in the introduction about the research and related references, which then would be followed through in the investigation and then revisited in the discussion and conclusion sections.

Response

7a. We revised the manuscript accordingly. Please refer to the response to Comment 1a.

Comment

7b. Mention of new references and new points of view should be avoided in the discussion and conclusion sections. Only points that have been worked with during the investigation and mentioned earlier, and which are part of the context of the research, should be included in the discussion and conclusion sections. Some of the material and references could be moved from the discussion section into the introduction. This would provide a better introduction to the study and make the line of thought more logical. Some of those points could be revisited in the discussion section.

Response

7b. We revised the manuscript accordingly. Please refer to the response to Comments 1b and 5.

Reviewer 2 Report

The text is very interesting and academically valuable. I consider the results to be enriching for the subject. There are small details that would improve it:

1- You have to read it in full because there are Chinese characters in the text that I do not understand what they refer to or if it is an editing error.

2-They need to make it clear why if the study was done in 2014, it came to an article until 2022. This is an important issue because it can lead to the validity or originality of the results being questioned.

3-I suggest that in the demographic explanation of the sample, put a table. Long paragraphs with so much data are confusing to read.

4-Care should be taken to ensure that the wording is objective in terms of the subject matter at all times. In some parts you read value judgements that show a certain subjective tendency.

5- The most delicate point that MUST be corrected are the references used to support the theoretical framework and the article in general. Only 25% of the texts used are from the last 5 years. This is very delicate for a subject as current as the one raised in the article, as it raises the question of whether it is indeed new or original or simply whether adequate research was not carried out beforehand. This MUST be corrected.

Overall, in all other respects, the article is interesting and valuable.

Author Response

We appreciated your valuable comments. As discussed below, we have revised our manuscript with underlines based on your suggestions. Please let us know if we need to provide anything else regarding this revision.

Comment

  • You have to read it in full because there are Chinese characters in the text that I do not understand what they refer to or if it is an editing error.

Response

Thank you for your reminding. We rechecked the text and confirmed that there was no Chinese character appearing in the text. Please let us know if any error occurred in the process of changing versions.

Comment

  • They need to make it clear why if the study was done in 2014, it came to an article until 2022. This is an important issue because it can lead to the validity or originality of the results being questioned.

Response

Thank you for your comment. This study was conducted between August 2014 and July 2015. The manuscript examining the psychometric propensities of the Parental Smartphone Use Management Scale (PSUMS), the main measure used in this study was published in 2019. Then we prepared and submitted the present manuscript. However, the reviewing process was delayed by a journal for almost one year. Therefore, we submitted it to international Journal of Environmental Research and Public Health in 2022. We reviewed the studies published until 2022 July and found that there was no study examining the factors related to parental self-efficacy in managing adolescent smartphone use. Therefore, we believed that this manuscript could provide new knowledge to the field of adolescent smartphone use.

Comment

3-I suggest that in the demographic explanation of the sample, put a table. Long paragraphs with so much data are confusing to read.

Response

Thank you for your suggestion. We added the demographics into Table 1 and reduced the paragraph of introducing demographics. Please refer to Line 247 (Table 1).

Comment

4-Care should be taken to ensure that the wording is objective in terms of the subject matter at all times. In some parts you read value judgements that show a certain subjective tendency.

Response

Thank you for your reminding. We revised the contents that might show a subjective tendency into an objective pattern. We raised two new contents below as examples.

Multiple biopsychosocial characteristics may contribute to the high risk of problematic internet/smartphone use in adolescents with ADHD, including proneness to boredom, aversion to delayed rewards, difficulties in controlling impulsivity and social adaptation, and rapid habituation to repeated positive reinforcement [35]. Please refer to Line 47-51.

Adolescents with severe ODD symptoms often argue with people in authority and actively refuse to comply with parents’ requests. This defiance may increase parents’ difficulties in managing adolescents’ smartphone use behaviors. Please refer to Line 274-277.

Comment

5- The most delicate point that MUST be corrected are the references used to support the theoretical framework and the article in general. Only 25% of the texts used are from the last 5 years. This is very delicate for a subject as current as the one raised in the article, as it raises the question of whether it is indeed new or original or simply whether adequate research was not carried out beforehand. This MUST be corrected.

Response

Thank you for your comment. We added 12 references (references 2, 5, 7-14, 28 and 29) published in recent 5 years to support the originality of this study. We further confirmed that there has no similar study examining the parental self-efficacy in managing adolescent smartphone use and related factors.

Comment

Overall, in all other respects, the article is interesting and valuable.

Response

Thank you for your positive comment on our manuscript.

Round 2

Reviewer 1 Report

The literature review part of the manuscript is now more informative.

On the whole, the manuscript has been improved.

Minor points can be made for further improvements, such as:

 L. 69-71

“Parental efficacy in managing adolescent smartphone use (PEMASU) may affect whether parents can successfully carry out the management for adolescent problematic smartphone use; however, it has not being deeply discussed.”

 For this sentence, you may want to change the wording along those lines.

“Parental efficacy in managing adolescent smartphone use (PEMASU) may affect whether parents can successfully carry out the management for adolescent problematic smartphone use; however, this has not been discussed in depth in the literature.”

L. 317

4.3. Limitations and Directions of Further Study

I suggest changing the title of this section to:

4.3. Limitations and Further Study

Author Response

We appreciated your valuable comments. As discussed below, we have revised our manuscript with underlines based on your suggestions. Please let us know if we need to provide anything else regarding this revision.

Comment

Line 69-71:

“Parental efficacy in managing adolescent smartphone use (PEMASU) may affect whether parents can successfully carry out the management for adolescent problematic smartphone use; however, it has not being deeply discussed.”

For this sentence, you may want to change the wording along those lines.

“Parental efficacy in managing adolescent smartphone use (PEMASU) may affect whether parents can successfully carry out the management for adolescent problematic smartphone use; however, this has not been discussed in depth in the literature.”

Response

Thank you for your suggestion. We revised this sentence based on your suggestion. Please refer to line 71.

Comment

Line 317

4.3. Limitations and Directions of Further Study

I suggest changing the title of this section to:

4.3. Limitations and Further Study

Response

Thank you for your suggestion. We revised this sentence based on your suggestion. Please refer to line 316.